# The Risk Factors Associated with the Prevalence of Multimorbidity of Anaemia, Malaria, and Malnutrition among Children Aged 6–59 Months in Nigeria

**DOI:** 10.3390/ijerph21060765

**Published:** 2024-06-13

**Authors:** Phillips Edomwonyi Obasohan, Stephen J. Walters, Richard M. Jacques, Khaled Khatab

**Affiliations:** 1School of Medicine and Population Health, Division of Population Health, University of Sheffield, Sheffield S1 4AD, UK; s.j.walters@sheffield.ac.uk (S.J.W.); r.jacques@sheffield.ac.uk (R.M.J.); 2Department of Liberal Studies, College of Business and Administrative Studies, Niger State Polytechnic, Bida Campus, Bida 912231, Nigeria; 3Faculty of Health and Wellbeing, Sheffield Hallam University, Sheffield S10 2BP, UK; k.khatab@shu.ac.uk

**Keywords:** multiple diseases, anaemia, malaria fever, malnutrition, syndemic, childhood, sustainable development goals

## Abstract

In the last ten years, multimorbidity in children under the age of five years has become an emerging health issue in developing countries. The study of multimorbidity of anaemia, malaria, and malnutrition (MAMM) among children in Nigeria has not received significant attention. This study aims to investigate what risk factors are associated with the prevalence of multimorbidity among children aged 6 to 59 months in Nigeria. This study used two nationally representative cross-sectional surveys, the 2018 Nigeria Demographic and Health Survey and the 2018 National Human Development Report. A series of multilevel mixed-effect ordered logistic regression models were used to investigate the associations between child/parent/household variables (at level 1), community-related variables (at level 2) and area-related variables (at level 3), and the multimorbidity outcome (no disease, one disease only, two or more diseases). The results show that 48.3% (4917/10,184) of the sample of children aged 6–59 months display two or more of the disease outcomes. Being a female child, the maternal parent having completed higher education, the mother being anaemic, the household wealth quintile being in the richest category, the proportion of community wealth status being high, the region being in the south, and place of residence being rural were among the significant predictors of MAMM (*p* < 0.05). The prevalence of MAMM found in this study is unacceptably high. If suitable actions are not urgently taken, Nigeria’s ability to actualise SDG-3 will be in grave danger. Therefore, suitable policies are necessary to pave the way for the creation/development of integrated care models to ameliorate this problem.

## 1. Introduction

Childhood mortality and morbidity rates are still very high, especially in low- and middle-income countries (LMIC), and these issues have resulted in a severe public health burden [1]. According to the World Health Organization (WHO) and the Centres for Disease Control and Prevention (CDC), about 25% of the world’s population is anaemic, with expectant mothers and children under the age of five years being the most vulnerable [2,3,4], but since 2016, the prevalence of anaemia has increased globally by more than 40% annually [5]. Similarly, over the last twenty years, malaria has remained a primary public health concern [6], with over 300 million cases reported in 2018 [7]. It has remained a leading cause of morbidity and mortality with LMICs, especially Sub-Saharan Africa (SSA), and contributes more than 80% of the global malaria burden [8,9]. Despite the considerable global decline in childhood stunting, over 150 million, 50 million, and 38 million children remain stunted, wasted, and overweight, respectively [10]. However, in 2018, there were more than 40 million overweight children under the age of five years, which was contrary to expectations and in keeping with a global target regarding malnutrition to keep the rate of childhood obesity constant [11], indicating a gradual global rise in the number of overweight children. However, it becomes more worrisome when children simultaneously contract two or more of anaemia, malaria, and malnutrition (MAMM). Multimorbidity is a condition where an individual displays two or more diseases and is becoming an emergent health condition among children in low- and middle-income countries (LIMCs).

There is evidence in research that anaemia, malaria, and malnutrition interrelate, resulting in adverse health outcomes and mortality, especially in children under five years of age. The concurrent study of these three diseases outcome has been lacking partly due to a lack of data. The 2018 Nigeria Demographic and Health Survey (NDHS) marked the first time, to the best of the authors’ knowledge, that data on these three outcomes had been collected simultaneously on a national scale in Nigeria, and perhaps in SSA. In a recent study conducted on SSA, malnutrition is a vital factor causing a high proportion of malaria-related mortality [12]. Also, malaria is strongly related to anaemia in childhood [13,14]. The relationship between malaria and malnutrition has remained a topical issue Sakwe et al. [15] found a significant interrelationship between malnutrition and malaria. Likewise, due to malaria’s relationship with anaemia and malnutrition, Teh et al. [16] recommended that reasonable control of anaemia and malnutrition will require the adequate control of malaria infections. The anaemic child presents a more significant measure of undernutrition [15]. On the other hand, most important is the emergence of the coexistence of these disease conditions and many other illnesses in individuals. However, recent studies independently examining these three disease conditions, using the same dataset, found that child’s age, maternal educational status, household wealth status, and region of residence are some of their overlapping significant predictors [17,18,19]. Despite its complexity, the relationship between malaria, anaemia, and malnutrition is essential for our knowledge of childhood morbidity and the formulation of successful intervention methods [12].

To change the trajectory of providing adequate treatment for children living with multimorbidity, more research, with a developing-country focus, is urgently needed to understand the risk factors and aetiologies associated with multimorbidity in children. This study is a step in the right direction in unravelling the epidemiology and determinants of multimorbidity of common childhood diseases that will help government, policymakers and implementors in formulating integrated guidelines suitable for managing multiple occurrences of diseases in children. Therefore, the aim of this study was to investigate the associations between potential risk factors for multimorbidity of anaemia, malaria, and malnutrition (MAMM) among children aged 6–59 months in Nigeria using the 2018 Nigeria Demographic Health Survey (NDHS) and the 2018 National Human Development Report (NHDR) datasets.

## 2. Materials and Methods

### 2.1. Sources of Data

The 2018 Nigeria Demographic and Health Survey (NDHS) was the primary source of data evaluated in this study. A cluster-based, multi-stage stratified random sampling design was used for this survey. Men and women between the ages of 15 and 59 years were interviewed, together with information on their children under the age of five, to learn more about the socioeconomic, demographic, environmental, and health features of households in Nigeria. The second dataset was from the 2018 Nigerian Human Development Report (NHDR) which was designed to provide comprehensive indices of human development beyond the conventional indices of income and well-being. Data at the state level not collected in 2018 NDHS were extracted and merged with 2018 NDHS as contextual variables.

### 2.2. The Outcome Variables

Three outcome or response variables were considered in this study as indicators for a common childhood multimorbidity cluster: anaemia, malaria, and nutritional status. These health conditions were captured objectively at the individual level in accordance with the WHO’s recommended procedures. The classifications of anaemia and malaria have been described elsewhere [17,18]. However, a measure of the overall description of malnutrition among children aged 6–59 months in Nigeria was assessed using four indicators, stunting, wasting, underweight, and overweight. The idea of using these four indicators as proxies for malnutrition may be unusual because, in the past, researchers believed that overweight and indicators of undernutrition do not share common factors. This is no longer the case, as literature is replete with is evidence to suggest emergence of the double burden of malnutrition [20], and that stunting (an indicator of undernutrition) correlates with overweight in a population [21,22,23,24]. Therefore, children with no trace of anthropometric failure were classified as ‘0’, labelled as ‘well nourished’, and those that had at least one of the four indicators were classified as ‘1’, labelled as ‘poorly nourished’ [20,25,26,27].

To classify multimorbidity across anaemia, malaria, and malnutrition, the ‘Composite Index of Multi-morbidity’ (CIMM), a technique adapted from the Composite Index of Anthropometric Failure (CIAF), was used [26,27]. The multimorbidity (with three outcomes) was classified into eight independent groups (including ‘no disease status’), such that multi-categorical responses could be represented in three intersecting sets of anaemia, malaria, and malnutrition (Figure 1).

From the intersecting diseases (Figure 1), this method recognised four distinct groups/categories: those that had ‘no disease’ at all and were classified as ‘0’; those that had ‘one disease only’ and were classified as ‘1’; those that had two diseases only, classified as ‘2’; and those that had all three diseases, classified as ‘3’. However, to align with the definition of multimorbidity as the cooccurrence of two or more diseases in an individual, categories 2 and 3 were grouped into one (having two or more diseases). Therefore, the ordered set of multimorbidity conditions was S = {0, 1, 2}.

### 2.3. Risk Factors (Predictor Variables)

The risk factors considered for this study were categorised following earlier research, and as presented in the NDHS 2018 final report, the factors that affect a childhood’s risk of contracting multimorbidity of anaemia, malaria, and malnutrition (MAMM) were broken down into clusters of factors related to the child, the parents, the household, the community, and the area. The covariates accounted for in this study included child-related (sex, age, birth size, preceding birth interval, the child taking iron supplements, duration of breast feeding, the child having been dewormed, the child having had a fever, and the child’s place of delivery); parent-related (maternal highest education, the mother residing with a partner, the mother’s religious status, the mother’s anaemia status, paternal work status, and paternal education status); household-related (household wealth status, the under-five-year-old child sleeping under a bed net, the sex of the household head, and the number of persons in the household); community-related (community wealth level, the proportion of those whose distance to a health facility is not problematic, and the community maternal education level); and state/area-related (state multidimensional poverty index, state human development index, state gender inequality index, state region of residence, and place of residence) variables. Furthermore, the definitions, classifications, and selection processes of these covariates have been reported elsewhere [17].

### 2.4. Analyses Techniques

#### Levels of Statistical Analyses

At the first level, percentage frequency counts and bivariate (Chi square) analysis were used to establish the prevalence and association of MAMM across some variables and the spatial map descriptions across the states and FCT and regions of residence in Nigeria. At the second level, a multivariate multilevel mixed-effect ordered logistic regression model was fitted to determine the multiple overlaps in the variables that simultaneously predict the number of occurrences of MAMM among children aged 6–59 months in Nigeria. Since there is no standard procedure to directly check the proportionality, assumptions using a multilevel mixed-effect ordered logistic regression model, a ‘naïve’ approach was adopted as recommended in [28], and the steps were as follows:Based on the full model (considering all the independent variables), the *Brant* test was used after performing ordinal logistic regression to identify which variables were proportional to their coefficients or not. Then, the individual predicted probabilities and the mean ${(\bar{x}}_{1})\;$ were computed.The *Brant* test was also performed based on the partial model (after removing those variables that violated the proportionality assumption in step 1) and found no more violation of the proportionality assumption. Then, individual predicted probabilities and the mean ${(\bar{x}}_{2})$ were computed.The test of difference in the two predicted means $\left({\bar{x}}_{1}-{\bar{x}}_{2}\right)$ was performed and no significant difference at the 5% level (both at two- and at one-tailed test) was found.

Given the above, the assumption of non-violation of proportionality was upheld while using the mixed-effect ordered logistic regression analysis method to investigate the multiple overlaps in the association between the individual factors, contextual factors, and MAMM among children 6–59 months of age in Nigeria. Nevertheless, in some situations, the issue of violation of assumptions may not cause serious problems, especially when the sample size is large [28].

The missingness observed in the selected variables ranges from 3 to 1610. Therefore, the multiple imputation method was used to replace missing values in the data set. All of the analyses were performed using Stata MP4 version 17 (Stata Corp, College Station, TX, USA) at a 5% alpha level of statistical significance.

## 3. Results

The multimorbidity of childhood diseases was assessed using the counts of the interactions between the three outcome variables. The mean number of diseases displayed by the children was 1.5, with a standard deviation of 0.96. Figure 2 shows the distribution of the three diseases’ possible interactions and their counts. There were more children displayed anaemia only, 22.5% (2293/10,183), compared with malaria only, 3% (308/10,183), and malnutrition only, 9% (897/10,183). Almost equally as many children, 16.9% (1721/10,183) and 17.4% (1767/10,183 display ‘all three disorders’ and ‘do not have any of the three diseases’.

### 3.1. Characteristics Associated with the Prevalence of Multimorbidity

Table 1 presents the distributions and the associations of individual and contextual variables with MAMM among children aged 6–59 months in Nigeria. Overall, 17.3% (1767/10,184) of the children had none of MAMM, 34.4% (3499/10,184) displayed only one of the diseases, while 48.3% (4918/10,184) had two or more of MAMM. Also, the proportion of children with two or more diseases was higher for male children, 50.62% (2641/5217), compared to their female counterparts, 45.8% (2277/4967). In addition, children with an average birth size, 78.7% (7914/10,059), were more commonly represented in the sample, such that the proportion of those with multimorbidity was highest among the small birth size group, 55.1% (673/1222), followed by those with an average birth size, 47.8% (3785/7914).

Furthermore, in the household wealth quintiles, the highest proportion of multimorbidity was among children from the poorest households, 73.6% (1393/1893). Additionally, the proportion of children with multimorbidity from a household containing four to six people was the lowest, 42.3% (2048/4836), compared to other households in the survey. Children from communities whose household wealth level was below the median (high) displayed a greater prevalence of multimorbidity, 66.4% (3086/4647). Moreover, the proportion of children displaying multimorbidity from a community with ‘lower than the median proportion of community maternal education level’ was 63.6% (3194/5025). In addition, the proportion of children with multimorbidity from urban areas, 34.3% (1538/4483), was lower than their counterparts in rural areas, 59.3% (3379/5700).

#### 3.1.1. Spatial Proportions of the Multimorbidity of Two or More Diseases by State and Region

Figure 3a reveals that the proportion of children displaying ‘two or more diseases’ in Nigeria was highest in Kebbi state, with 0.83 (95% CI: 0.78–0.86), followed by Jigawa state, 0.73 (95% CI: 0.69–0.78).

Similarly, Figure 3b showed that the proportion of children displaying ‘two or more diseases’ out of anaemia, malaria, and malnutrition was the highest in the northwest, 0.65 (95% CI: 0.62–0.67), followed by northeast geopolitical zone, with a proportion of 0.59 (95% CI: 0.56–0.62) of children living with MAMM. All of the geopolitical zones in the southern part of Nigeria had similar distributions of multimorbidity and had proportions below the national average of 0.48 (95% CI: 0.47–0.49), including the north-central region, with a proportion of 0.44 (95% CI: 0.41–0.47). Moreover, comparing the two maps, all of the states in the north-central region and states in the extreme northwest have similar proportions, while states in the northeast showed some levels of variation.

#### 3.1.2. Multivariate Multilevel Analysis of Multimorbidity Status

Based on a multicollinearity check using the combination of backward and forward stepwise selection, 28 variables were finally included in the analysis. The definitions and classifications of these variables have been reported elsewhere [17].

#### 3.1.3. Multilevel Mixed-Effect Ordinal Logistic Regression Models

The proportionality assumption was checked through a naïve method, showing that it was not violated. The predicted mean multimorbidity of the full model was 0.170, while the predicted mean multimorbidity of the partial model (after the variables that violated the proportionality assumptions were removed) was 0.166. The null hypothesis that the difference between these two means was not significantly different from zero at *p* < 0.05 was not rejected (*p* = 0.0729). So, it can be concluded that the proportionality assumption was not violated. Therefore, three-level multilevel mixed-effect ordinal logistic regression was used in the analysis. In the first instance, it was confirmed that the two-level model was significantly nested within a three-level model using a likelihood ratio test with χ^2^ = 293.65, *p* < 0.0001.

### 3.2. Model Building

In this section, four models were evaluated. This study used a weighted sample size of 10,451 children at level 1, nested in 1378 communities at level 2, with an average of 8 children per community, in turn nested in 37 states at level 3, with an average of 283 children per state.

Model 0 contained no covariate (variance component model), while models 1, 2, and 3 contained predictors for levels 1, 1 and 2, and 1, 2 and 3, respectively. However, model 3, which contained all of the level 1, 2, and 3 variables (full model), was adjudged the model of best fit with the highest log-likelihood (−9189.8) and lowest AIC (18,519.6), and therefore was reported. The likelihood ratio test (1558.3), *p* < 0.0001, of the variance component model indicates that the multilevel mixed-effect ordinal logistic model is better than a single-level ordinal logistic model (See Table 2).

### 3.3. Fixed and Variance Effects of Multimorbidity

Table 2 reports the significance of the risk factors or predictors in the chosen model 3 (level 1, 2, and 3 covariates). It shows that the proportional odds of having MAMM in female children is 0.73 (95% CI: 0.67–0.80) times lower than the odds of male children after adjustment for the other variables in the model. Likewise, the proportional odds of children aged 12–23 months having two or more diseases versus a combination of zero and one diseases is 1.39 (95% CI: 1.18–1.65) times higher when compared to children aged 6–11 months with all other variables kept constant.

Furthermore, the results reveal that the child’s birth size, pre-birth interval, the child having been dewormed in the last six months before the survey, the child having had a fever in the last two weeks before the survey, child being delivered in a private health facility were significant factors. In addition, the maternal education status, religious status, anaemia status, body weight index status, paternal education status, the household wealth status, the proportion of people with higher than median wealth, the median proportion of people in the community who said that their distance to a health facility is not problematic were significant predictors. But, the community maternal education being ‘above the median’ was at borderline significance. However, among the state-level variables, the result shows that state multidimensional poverty index (SMPI), state human development index (HDI), and state gender inequality index (GII) were not significant. However, the region of residence and place of residence were significant predictors of the proportional odds of a child displaying MAMM versus the combination of zero and one disease among children 6–59 months of age in Nigeria compared with their respective reference category.

The variance components in Table 2 show that the intrastate and intracommunity correlations drop from 0.10 and 0.28 in the null model (model 0) to 0.005 and 0.09 in the model containing all 28 covariates (model 3), respectively. Also, the between-state and between-community variabilities drop from 0.10 and 0.18 in the null model to 0.004 and 0.08 in the full-level model (model 3), respectively. Again, this indicates that the distribution of all of the variables across states and communities differs significantly. Additionally, a measure of odds for cluster variance was computed using the median odds ratios (Table 2). For the choice model (model 3), the median odds ratio (MOR) computed for states was 1.14, 95% CI (1.10–1.24), signifying a 14% increased risk of a child contracting two or more diseases if he/she moves from one state to another with an increased risk of multimorbidity. Similarly, there is a 69% increased risk of a child contracting two or more diseases if he/she moves to another community with a higher risk of MAMM.

## 4. Discussion

The aim of this study was to investigate the associations between potential risk factors for multimorbidity of anaemia, malaria, and malnutrition (MAMM) among children aged 6–59 months in Nigeria using the 2018 Nigeria Demographic and Health Survey (NDHS) and the 2018 National Human Development Report (NHDR) datasets. A three-state multilevel mixed-effect ordinal logistic regression was used to find the significant predictors of displaying two or more diseases among children aged 6–59 months in Nigeria (using anaemia, malaria, and malnutrition as proxies for disease cooccurrence). Multimorbidity studies in children have been neglected for a long time, especially in LMICs. It is challenging to evaluate this study’s findings in light of earlier research, partly because of the lack of similar studies and differences in methodological approach, disease conditions, survey types, and population settings [29]. Nevertheless, this study finds significant disparities in some child-, parental-, household-, community-, and area-related risk factors in relation to the occurrence of MAMM.

This study found approximately a two-fold higher prevalence (48.3%) of MAMM among children aged 6–59 months in Nigeria than those living with one disease out of anaemia, malaria, and malnutrition, or none of the diseases [19]. Furthermore, the number of children with all three disease conditions was equally as high as the number of those that did not have any of the disease conditions. This finding supports the assertion that the three diseases coexist and may cluster among children in Nigeria.

Similarly, spatial disparities in the proportion of children were found across the regions and states in Nigeria. This proportion was highest in Kebbi state, followed by Jigawa state. Also, the northwest region had the highest proportion of children with MAMM compared to the other regions of residence in Nigeria. The likely reason for this is the high rate of insecurity in the region for almost a decade and half, leaving the people, especially women and children, homeless (living in internally displaced person’s camps). These people cannot carry out farming activities, so they do not have easy access to good food and drinking water.

The findings show that child’s sex is significantly protective for female children against MAMM compared to their male counterparts. This finding is supported by previous studies, which revealed that female children are less likely to display multimorbidity of childhood diseases [30,31], but is contrary to Tran et al. [32], who concluded that it was more likely for a female to display two or more diseases. In the past (pre-Millennium Development Goals era), gender inequalities were seen to pose a disadvantage for female children [33]. However, in recent studies [34], and across the various disease spectra, female children were at an advantage. The possible reason for this is that in some cultures, female children are more home-based and closer to their mothers as they grow up, so their welfare is more often taken care off, while male children play more outdoors, scavenging around the slums and environs. Similarly, the older the children become, the more likely they are to display MAMM. Children aged between 1 and 3 years are at significantly increased risk of displaying MAMM compared to children aged less than 1 year. This finding is supported by the study conducted in [35] in the same country, and elsewhere in [31,32]. The possible reason for this is that these children are transitioning from being weaned off breastfeeding to supplementary feeding. Most children at this stage would prefer to continue with breast milk rather than eat any other food being introduced by their mothers. However, contrary results that being older is protective for children under five years of age compared with children aged less than one year were reported [21,30,36]. The possible reasons for this might be differences in disease composition, survey type, and research location. Additionally, the findings in this study also indicated that children born with small size were more likely to display MAMM than large birth-sized children. This finding does not support the conclusion made by [35], who reported that children born large birth-size are likely to have an increased risk of multimorbidity of fever, pneumonia, and diarrhoea. These contrary findings may be because these two studies used different disease conditions as a proxy for multimorbidity. There are three primary reasons that a child may be born with a low birth weight (LBW)—this could be due to genetics because the parents are small, intrauterine growth restriction (IUGR) [37], or because the baby is born pre-term (before 37 weeks of pregnancy) [38].

The findings also show that maternal education has a negative correlation with children’s health outcomes [35]. Moreover, the children of mothers with higher educational status had a lower risk of multimorbidity—defined as having two or more of anaemia, malaria, and malnutrition—compared with children of mothers with no formal education, conditional upon holding other predictors constant. This finding is consistent with the conclusions of another research [31,32]. The reason for is that these mothers were likely to have more robust knowledge about health care to safeguard their children and handle these illnesses, better [39].

Additionally, our findings show that children whose mothers were anaemic are at a higher risk of displaying two or more out of MAMM. The possible explanation from previous studies is that children of anaemic mothers are more likely to be anaemic and are at high risk of poor child health outcomes [18,40]. The mechanism through which maternal anaemia status can affect children 6–59 months of age is complex. More important are the causes of maternal anaemia, especially during pregnancy, which include poor diet (deficiencies in iron, folic acid, and vitamins), infections like malaria, and untreated hereditary haemoglobin abnormalities [40,41].

From the group of household-related predictors, only household wealth quintile had significant effects on childhood MAMM when other predictors are held constant. A higher household socioeconomic level was associated with a lower risk of developing two or more out of MAMM. This finding agrees with the conclusions of [32,35,36]. However, children from lower-income families are more likely to contract childhood diseases. This result relates to the fact that low-income families may find it challenging to purchase nutritious food that will increase their intake of nutrients and help them develop immunity to diseases [35]. They may also live in impoverished or densely populated areas [42] and may have more children than they can reasonably care for [43], which is likely to increase the risk of multimorbidity in children.

This current study found that the higher (median and above) the proportion of community wealth status, the more protected the children are from MAMM. This finding supported other studies indicating that children from low-income households or communities have poor health outcomes [44]. Similarly, this study reports that the higher the proportion of those in the community who affirmed that the distance to the nearest health centre is not problematic, the less likely it is that the children from such a community will contract two or more out of MAMM. The distance to health centres could be a significant factor in receiving prompt medical attention, and delays in getting treatment for paediatric illnesses can have more severe consequences. Oldenburg et al. agree that children that have less access to primary care may be more susceptible to poor health outcomes, including mortality; as a result, this may represent a population for which initiatives aimed at reducing child mortality and morbidity should be prioritised [45]. Some studies disagree with this finding [46,47].

Based on the area-related predictors, this current study found that children from states with a high human development index were significantly more likely to display MAMM than those living in states with the lowest HDI. This finding is contrary to the general expectation that the higher the state HDI, the less likely it is that the children from such a state will display adverse health outcomes [48]. The possible explanation for this result is that 14 out of 37 states were classified as ‘high HDI’, leading to high variability in MAMM in this group, compared with six states classified as ‘lowest HDI’, with low variability in MAMM. The high heterogeneity in MAMM across these 14 states must have resulted in their weak effects and significance. So, a non-linear impact of HDI on a child’s health outcomes is therefore suspected [49]. The reasons for this happening can be explored further in future studies.

Finally, the place of residence was a significant predictor of MAMM. The results from this study show that children in rural areas of Nigeria were at increased risk of displaying MAMM when all other covariates were kept constant. This finding did not agree with another study [32]

## 5. Strengths and Limitations of the Study

The dataset from the 2018 NDHS, which included merged contextual characteristics from the 2018 NHDR, was used in this study. Anaemia, malaria, and malnutrition are three objectively assessed (standard WHO measurement procedures) paediatric diseases combined for the first time in the DHS data collection. So, to the best of the researchers’ knowledge, this study represents the first joint modelling of MAMM among children aged 6–59 months in Nigeria or elsewhere undertaken on the national scale in SSA. Since this was a baseline study, it employed a more simplified approach by developing a composite score that indicated the order in which the combination of these diseases occurred based on the generally accepted definition of multimorbidity as the cooccurrence of two or more diseases in an individual without reference to an index disease.

Also, the dataset came from a nationally representative survey with abundant evidence of hierarchy. Yet, most previous studies did not account for the multilevel structure or use the proper statistical techniques. This study applied multilevel methods to account for individual, community, and state variations.

However, this study is not without its limitations. First, the survey being cross-sectional meant that this study could only examine the associations between variables. Therefore, causality could not be ascertained [34,35,50]. Secondly, this study assumed that the three diseases were of equal importance or severity (e.g., having malaria was the same as having anaemia or malnutrition). Thirdly, only Nigeria was the subject of this study, which may limit its capacity to be generalised to other SSA countries. In view of the interrelationships between the three disease conditions, endogeneity was ignored in this investigation; therefore, we might have certified some predictors as significant when they could have just as easily been due to chance [51]. The wrong assumptions may therefore have been drawn. In addition, only individual-level weighting was considered. These limitations are suitable for future investigation.

## 6. Policy Implications

This study found that, at the time of the survey, about one in every two (48.3%) Nigerian children aged 6–59 months displayed two or more diseases out of anaemia, malaria, and malnutrition, while one in every six (16.9%) children suffered from all three diseases simultaneously. These findings are alarming on a national scale, and this will likely be the case for children across sub-Saharan Africa (SSA). This situation requires urgent policies and a coordinated approach to avoid further escalation, because these three conditions individually contribute to high mortality in children in LMICs, and among children displaying all three diseases at the same time, mortality will be much higher. In effect, this result shows that the situation of multimorbidity among children aged 6–59 months in Nigeria could worsen if nothing is done to reduce its prevalence, therefore making it impossible for Nigeria to attain SDG-3 by 2030. Some areas requiring urgent attention are as follows:These spatial map descriptions of the prevalence of MAMM could be used by decision-makers to quickly target development efforts for health care, food security and poverty alleviation interventions in areas with a high prevalence of MAMM. Hence, success-driven efforts to end the long-lasting security issues in these regions are required so that people already displaced can return to their homes to farm again.Addressing the disparities between genders in childhood MAMM in Nigerian community, mobilisation based on gender-based prejudice and gender-sensitive policies is needed.

Nutrient-fortified meals can be given to children in this age group, helping to sustain their immunity from their mother’s breast milk until their natural immunity is built up at a later age. To detect issues with foetal growth, antenatal care is crucial and strongly encouraged for every pregnant woman. The ongoing reforms in Nigeria’s health sector aim to dramatically alter the way in which health care is delivered throughout Nigeria by increasing the accessibility, quality, and availability and reducing the cost of health care services via promoting private sector investment and engagement [52], making primary health care services accessible at the grassroots level, with women and children being the foremost priority. Therefore, it is strongly advised that would-be mothers should adequately space their children at intervals of over 24 months. This recommendation is because mothers who have their next child after waiting for at least two years can recover most of their body’s nutrients and blood lost during the first pregnancy and breastfeeding [53].

A deworming program that includes children under five should also be initiated. On the other hand, viral and bacterial infections are major causes of non-malaria fever among a significant number of children in Nigeria [54,55]. Therefore, as part of health education at antenatal clinics, it is crucial to stress the importance of personal and environmental cleanliness initiatives. Given this, girls’ education should be encouraged, especially in the northern part of the country, where MAMM is highly prevalent. Better education for mothers, particularly caregivers, often improved their knowledge, attitude, and practice of typical children’s diseases [17].

Public health initiatives to lower childhood MAMM should pay more attention to reducing poor nutrition and other infections among pregnant, nursing mothers and children still breastfeeding. For instance, between 2013 and 2025, the Nigerian National Policy on Food and Nutrition has as one of its goals the reduction in maternal anaemia during pregnancy by 27% [18,56]. Therefore, there should be political commitment to make this happen. However, the high rate of poverty among families in Nigeria should be addressed. This provision will enable families to afford good nutrition sources. To address this situation, social security that will take care of the immediate necessities of life (food, shelter, and health) for poor households in Nigeria should be in place.Also, public health measures to lower childhood MAMM should pay more attention to increasing access to health centres through expanding the primary health care system.Finally, public health approaches should include making health care facilities closer to people in rural areas. More importantly, there should be an assurance that health workers posted to these rural areas are well remunerated and monitored in order to dwell among these people.

## 7. Conclusions

This study is the first of its kind to comprehensively examine the risk factors for the prevalence of MAMM among children aged 6–59 months in Nigeria, for which nationally representative data have been made available for the first time. It serves as a baseline for other studies to build on. This study’s objectives include determining the prevalence of MAMM across states and regions in Nigeria. According to this study, two or more diseases are present in almost half of Nigerian children aged 6 to 59 months. In addition, 17% of the children in the sample were simultaneously living with malaria, anaemia, and malnutrition. This is increasingly worrying, providing an indication that severe inequalities in multimorbidity of childhood health outcomes could cut across individual and contextual characteristics of children aged 6–59 months in Nigeria and remain unresearched. Funding bodies should show more interest in funding research on childhood multimorbidity in developing countries. Furthermore, although longitudinal studies on children in LMICs may be difficult, they are required to demonstrate the importance of the predictors of MAMM and identify causal relationships over time. When longitudinal data become available for these diseases, a more sophisticated modelling approach on the connection between anaemia, malaria, and malnutrition can be developed to enhance integrated care for children beyond how it is currently treated. Therefore, this study has exposed the need for urgent responses through creating and executing good policies to address these situations if SDG-3 is to be realized in Nigeria. In addition, the results have demonstrated the need for clinicians and health care providers to develop integrated care models suitable for managing and treating children displaying multiple diseases (especially anaemia, malaria, and malnutrition). There is an urgent need for research in multiple diseases that will lead to a paradigm shift in the training curriculum of medical schools from the clinical guidelines for treating single diseases, as is presently the case, to include handling clusters of diseases [57], especially among children in LMICs. So, a deeper understanding of these issues around MAMM might result in the creation of management guidelines that allow patients to obtain more effective and efficient care.

## Figures and Tables

**Figure 1 ijerph-21-00765-f001:**
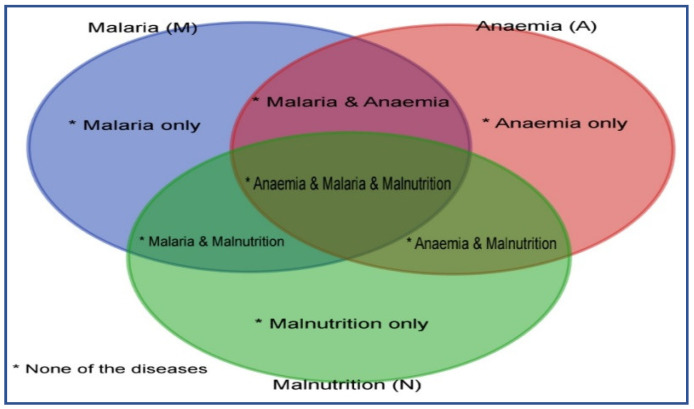
Diagram representing the intersection of the three outcome diseases.

**Figure 2 ijerph-21-00765-f002:**
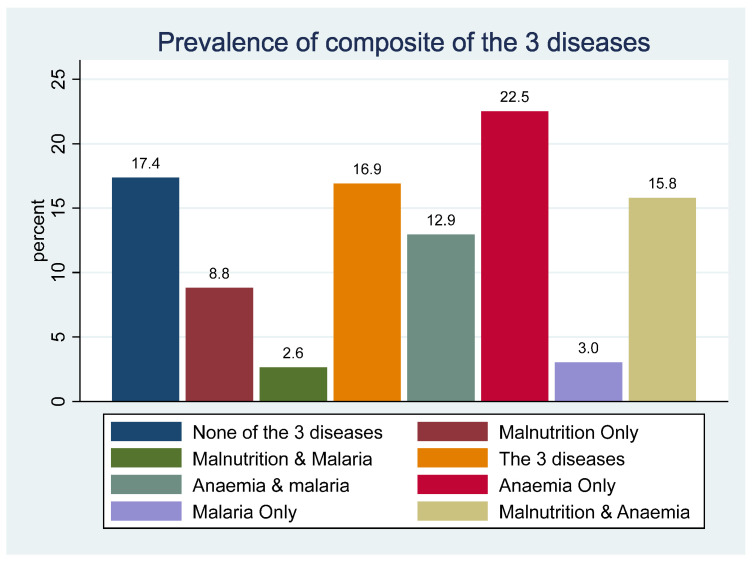
Distribution of prevalence of combinations of the 3 diseases.

**Figure 3 ijerph-21-00765-f003:**
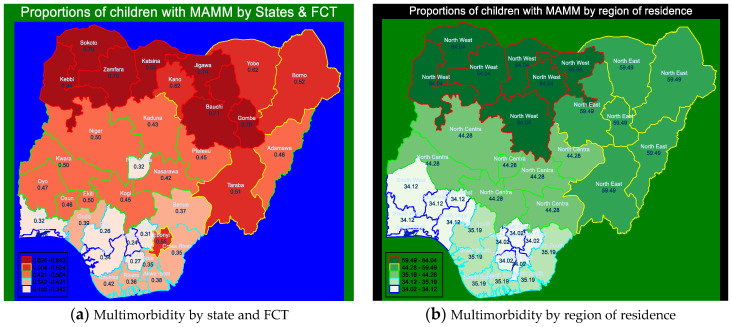
Spatial maps describing the proportions of children with two or more diseases. Source: Data computed from Nigeria DHS 2018.

**Table 1 ijerph-21-00765-t001:** Individual and contextual characteristics associated with multimorbidity among children aged 6–59 months in Nigeria (N = 10,184).

		Multimorbidity Status		
Variables	Total	None of the Diseases	One Disease Only	Two or More of the Diseases
	N (%)	N (%)	N (%)	N (%)
Prevalence rate		1767 (17.3)	3499 (34.4)	4918 (48.3)
Child-related characteristics				
Child’s sex		χ^2^ (2) = 25.03, *p* = 0.0002		
Male	5217 (51.23)	841 (16.13)	1734 (33.25)	2641 (50.62)
Female	4967 (48.77)	926 (18.64)	1764 (35.52)	2277 (45.84)
Child’s age in group		χ^2^ (8) = 205.55, *p* < 0.0001		
6–11 months	1232 (12.1)	165 (13.35)	566 (45.91)	502 (40.74)
12–23 months	2422 (23.78)	289 (11.95)	888 (36.66)	1245 (51.39)
24–35 months	2160 (21.21)	363 (16.82)	694 (32.13)	1102 (51.05)
36–47 months	2227 (21.87)	452 (20.32)	664 (29.83)	1110 (49.85)
48–59 months	2143 (21.04)	498 (23.23)	687 (32.06)	958 (44.71)
Child’s birth size		χ^2^ (4) = 38.28, *p* < 0.0001		
Large	923 (9.18)	190 (20.63)	343 (37.13)	390 (42.23)
Average	7914 (78.68)	1387 (17.53)	2741 (34.64)	3785 (47.83)
Small	1222 (12.15)	177 (14.52)	371 (30.37)	673 (55.11)
Preceding birth interval		χ^2^ (8) = 135.23, *p* < 0.0001		
None	1944 (19.13)	449 (23.09)	689 (35.42)	807 (41.49)
8–24 months	2191 (21.55)	319 (14.56)	736 (33.58)	1136 (51.87)
25–35 months	2884 (28.38)	435 (15.07)	924 (32.05)	1525 (52.88)
36–59 months	2351 (23.13)	383 (16.28)	827 (35.16)	1141 (48.55)
60+ months	795 (7.82)	177 (22.23)	317 (39.88)	301 (37.89)
Took Iron supplements		χ^2^ (2) = 67.80, *p* < 0.0001		
No	8224 (81.02)	1347 (16.38)	2748 (33.42)	4128 (50.2)
Yes	1927 (18.98)	416 (21.62)	737 (38.25)	773 (40.13)
Duration of breastfeeding		χ^2^ (4) = 119.28, *p* < 0.0001		
Ever breastfed, not currently breastfeeding	7440 (73.06)	1474 (19.82)	2448 (32.9)	3518 (47.29)
Never breastfed	171 (1.68)	19 (11)	68 (39.89)	84 (49.11)
Still breastfeeding	2572 (25.26)	274 (10.65)	983 (38.22)	1315 (51.13)
Child took deworming drug in the last 6 months		χ^2^ (2) = 348.84, *p* < 0.0001		
No	7235 (71.41)	1033 (14.28)	2302 (31.82)	3899 (53.89)
Yes	2897 (28.59)	729 (25.16)	1173 (40.5)	995 (34.33)
Total	10,132 (100)	1762 (17.39)	3476 (34.31)	4894 (48.3)
Child had a fever in last 2 weeks before the survey		χ^2^ (2) = 281.15, *p* < 0.0001		
No	7485 (73.52)	1473 (19.68)	2764 (36.93)	3248 (43.39)
Yes	2696 (26.48)	294 (10.9)	735 (27.25)	1667 (61.85)
Place of child’s delivery		χ^2^ (6) = 794.52, *p* < 0.0001		
Home	5348 (52.51)	572 (10.69)	1543 (28.86)	3233 (60.45)
Public facility	2975 (29.22)	672 (22.58)	1163 (39.08)	1141 (38.34)
Private facility	1660 (16.3)	490 (29.52)	701 (42.25)	469 (28.23)
Elsewhere	200 (1.97)	34 (16.83)	91 (45.59)	75 (37.59)
Parental-related characteristics				
Maternal/caregiver’s highest educational level		χ^2^ (6) = 1417.75, *p* < 0.0001		
No education	3970 (38.98)	332 (8.37)	951 (23.95)	2687 (67.68)
Primary	1643 (16.14)	221 (13.45)	581 (35.35)	841 (51.2)
Secondary	3597 (35.32)	832 (23.13)	1542 (42.88)	1223 (34)
Higher	974 (9.56)	382 (39.23)	425 (43.64)	167 (17.13)
Mother is currently residing with husband/partner		χ^2^ (2) = 20.09, *p* = 0.0022		
Living with her partner	8861 (91.05)	1479 (16.69)	3039 (34.3)	4343 (49.02)
Staying elsewhere	871 (8.95)	195 (22.41)	298 (34.22)	378 (43.37)
Mother’s religious status		χ^2^ (6) = 595.20, *p* < 0.0001		
Catholic	1027 (10.09)	241 (23.43)	442 (43.03)	344 (33.54)
Other Christian	3438 (33.76)	835 (24.28)	1378 (40.09)	1225 (35.63)
Islam	5654 (55.52)	682 (12.07)	1651 (29.2)	3321 (58.73)
Traditionalist and others	64 (0.63)	9 (14.57)	28 (42.81)	27 (42.62)
Maternal ethnicity		χ^2^ (6) = 737.24, *p* < 0.0001		
Hausa/Fulani/Kanuri/Seribiri	4067 (39.94)	407 (10)	1070 (26.3)	2591 (63.69)
Ibo	1650 (16.2)	404 (24.49)	710 (43.04)	536 (32.47)
Yoruba	1488 (14.62)	392 (26.34)	562 (37.73)	535 (35.93)
Others	2978 (29.24)	564 (18.94)	1157 (38.86)	1256 (42.19)
Mother’s anaemia status		χ^2^ (2) = 269.34, *p* < 0.0001		
Not anaemic	4206 (41.84)	997 (23.7)	1534 (36.47)	1675 (39.83)
Anaemic	5847 (58.16)	761 (13.02)	1930 (33.01)	3156 (53.97)
Maternal body weight status		χ^2^ (6) = 518.14, *p* < 0.0001		
Normal	5311 (60.82)	776 (14.62)	1757 (33.09)	2777 (52.29)
Underweight	885 (10.13)	84 (9.48)	233 (26.35)	568 (64.17)
Overweight	1668 (19.1)	429 (25.72)	689 (41.3)	550 (32.98)
Obese	869 (9.95)	255 (29.39)	405 (46.62)	208 (23.99)
Paternal work status		χ^2^ (2) = 6.04, *p* = 0.2041		
No	304 (2.98)	37 (12.09)	112 (36.88)	155 (51.02)
Yes	9880 (97.01)	1731 (17.52)	3387 (34.28)	4763 (48.21)
Partner education status		χ^2^ (6) = 969.76, *p* < 0.0001		
No education	2872 (29.91)	231 (8.04)	651 (22.66)	1990 (69.3)
Primary education	1423 (14.82)	183 (12.84)	485 (34.1)	755 (53.06)
Secondary education	3741 (38.95)	775 (20.72)	1514 (40.47)	1452 (38.81)
Tertiary education	1566 (16.31)	471 (30.1)	651 (41.55)	444 (28.35)
Household-related characteristics				
Household wealth index		χ^2^ (8) = 1635.53, *p* < 0.0001		
Poorest	1893 (18.59)	109 (5.73)	392 (20.7)	1393 (73.57)
Poorer	1989 (19.53)	166 (8.33)	555 (27.9)	1268 (63.77)
Middle	2139 (21)	328 (15.35)	753 (35.19)	1058 (49.46)
Richer	2144 (21.05)	445 (20.75)	876 (40.85)	823 (38.4)
Richest	2019 (19.83)	720 (35.66)	924 (45.75)	376 (18.6)
Children under 5 slept under a mosquito bed net last night		χ^2^ (6) = 176.44, *p* < 0.0001		
No child	1316 (13.02)	250 (19)	451 (34.24)	615 (46.75)
All children	4715 (46.64)	744 (15.78)	1549 (32.86)	2422 (51.37)
Some children	996 (9.85)	114 (11.42)	269 (26.97)	613 (61.61)
No net in household	3083 (30.5)	635 (20.6)	1211 (39.27)	1238 (40.13)
Sex of household head		χ^2^ (2) = 20.61, *p* = 0.0007		
Male	9096 (89.32)	1528 (16.8)	3124 (34.34)	4444 (48.85)
Female	1087 (10.67)	239 (21.98)	375 (34.46)	473 (43.56)
Number of people in household		χ^2^ (6) = 247.76, *p* < 0.0001		
0–3	979 (9.61)	195 (19.95)	354 (36.18)	429 (43.86)
4–6	4836 (47.48)	985 (20.37)	1802 (37.27)	2048 (42.36)
7–9	2462 (24.17)	372 (15.1)	842 (34.21)	1248 (50.69)
10+	1907 (18.73)	215 (11.28)	500 (26.22)	1192 (62.5)
Community-related characteristics				
Proportion of community wealth level		χ^2^ (2) = 1183.30, *p* < 0.0001		
Low	4647 (45.63)	387 (8.34)	1174 (25.26)	3086 (66.4)
High	5536 (54.36)	1380 (24.92)	2325 (41.99)	1832 (33.08)
Community distance to health facility is not problematic		χ^2^ (2) = 245.38, *p* < 0.0001		
Low	4702 (46.17)	629 (13.38)	1417 (30.14)	2656 (56.48)
High	5481 (53.82)	1138 (20.76)	2082 (37.98)	2262 (41.26)
Proportion of community maternal education level		χ^2^ (2) = 972.85, *p* < 0.0001		
Low	5025 (49.34)	494 (9.83)	1337 (26.6)	3194 (63.56)
High	5158 (50.65)	1273 (24.68)	2162 (41.91)	1723 (33.41)
Area-related characteristics				
Multidimensional Poverty Index by State (MPI)		χ^2^ (8) = 913.28, *p* < 0.0001		
Highly Deprived	847 (8.32)	57 (6.7)	208 (24.54)	582 (68.75)
Above averagely deprived	3093 (30.37)	309 (9.98)	791 (25.59)	1992 (64.43)
Averagely Deprived	2319 (22.77)	402 (17.36)	884 (38.14)	1032 (44.51)
Mildly Deprived	1939 (19.04)	487 (25.11)	720 (37.16)	732 (37.73)
Lowest Deprived	1987 (19.51)	512 (25.8)	895 (45.04)	579 (29.16)
Human Development Index by State (HDI)		χ^2^ (8) = 860.46, *p* < 0.0001		
Lowest HDI	2150 (21.11)	201 (9.35)	552 (25.68)	1397 (64.97)
Low HDI	2416 (23.73)	267 (11.06)	690 (28.56)	1459 (60.39)
Average HDI	2223 (21.83)	442 (19.9)	846 (38.08)	934 (42.02)
High HDI	2680 (26.31)	600 (22.37)	1096 (40.92)	984 (36.71)
Highest HDI	715 (7.02)	257 (35.97)	314 (43.88)	144 (20.15)
Gender Inequality Index by State (GII)		χ^2^ (8) = 551.09, *p* < 0.0001		
Lowest GII	2726 (26.77)	660 (24.23)	1129 (41.42)	936 (34.35)
Low GII	1171 (11.5)	266 (22.71)	507 (43.26)	398 (34.03)
Average GII	977 (9.59)	165 (16.91)	301 (30.79)	511 (52.3)
High GII	4054 (39.81)	557 (13.74)	1222 (30.13)	2275 (56.13)
Highest GII	1256 (12.33)	119 (9.46)	341 (27.13)	796 (63.41)
Region of residence		χ^2^ (10) = 761.25, *p* < 0.0001		
North-central	1436 (14.1)	277 (19.29)	523 (36.43)	636 (44.28)
North-east	1573 (15.44)	204 (13)	433 (27.5)	936 (59.49)
North-west	2967 (29.13)	286 (9.65)	781 (26.31)	1900 (64.04)
South-east	1328 (13.04)	292 (22.01)	584 (43.97)	452 (34.02)
South-south	1086 (10.66)	224 (20.61)	480 (44.21)	382 (35.19)
South-west	1793 (17.61)	483 (26.95)	698 (38.93)	612 (34.12)
Type of place of residence		χ^2^ (2) = 682.03, *p* < 0.0001		
Urban	4483 (44.02)	1117 (24.92)	1828 (40.77)	1538 (34.31)
Rural	5700 (55.97)	650 (11.4)	1671 (29.31)	3379 (59.28)

**Table 2 ijerph-21-00765-t002:** Multilevel ordinal logistic regression analysis of the individual-, community-, and state-level risk factors for MAMM.

	Model 0 (N = 10,451) No Covariates	Model 1 (N = 10,451) (Level 1 Covariates Only)			Model 2 (N = 10,451) (Level 1 and 2 Covariates Only)			Model 3 (N = 10,451) (Level 1, 2, and 3 Covariates)		
		aOR	*p*	95% CI	aOR	*p*	95% CI	aOR	*p*	95% CI
Child’s sex										
Male										
Female		0.76	<0.001	0.70–0.83	0.76	<0.001	0.70–0.83	0.76	<0.001	0.70–0.83
Child’s age in group										
6–11 months										
12–23 months		1.44	<0.001	1.23–1.68	1.43	<0.001	1.23–1.67	1.44	<0.001	1.23–1.68
24–35 months		1.34	<0.001	1.10–1.64	1.33	0.01	1.09–1.63	1.34	<0.001	1.10–1.63
36–47 months		1.11	0.31	0.91–1.36	1.11	0.33	0.90–1.35	1.11	0.30	0.91–1.36
48–59 months		0.89	0.27	0.73–1.09	0.89	0.24	0.72–1.08	0.89	0.25	0.72–1.09
Child’s birth size										
Large										
Average		1.09	0.22	0.95–1.26	1.10	0.19	0.95–1.27	1.09	0.23	0.95–1.26
Small		1.37	<0.001	1.14–1.65	1.38	<0.001	1.14–1.66	1.38	<0.001	1.14–1.66
Preceding birth interval										
None										
8–24 months		1.31	<0.001	1.14–1.50	1.31	<0.001	1.15–1.51	1.31	<0.001	1.15–1.50
25–35 months		1.11	0.13	0.95–1.26	1.11	0.11	0.98–1.27	1.12	0.10	0.98–1.27
36–59 months		1.01	0.90	0.88–1.15	1.02	0.82	0.89–1.16	1.02	0.79	0.89–1.17
60+ months		0.89	0.21	0.75–1.07	0.90	0.24	0.75–1.07	0.90	0.22	0.75–1.07
Child took Iron supplements										
No										
Yes		1.06	0.30	0.95–1.19	1.07	0.28	0.95–1.20	1.06	0.32	0.94–1.19
Duration of breastfeeding										
Ever breastfed, not currently breastfeeding										
Never breastfed		1.30	0.12	0.94–1.81	1.28	0.14	0.93–1.78	1.26	0.16	0.91–1.75
Still breastfeeding		0.97	0.70	0.83–1.13	0.96	0.64	0.83–1.12	0.97	0.66	0.83–1.13
Child was dewormed in last 6 months before the survey										
No										
Yes		0.80	<0.001	0.72–0.89	0.80	<0.001	0.72–0.89	0.81	<0.001	0.72–0.90
Child had fever in last 2 weeks before the survey										
No										
Yes		1.59	<0.001	1.43–1.75	1.58	<0.001	1.43–1.75	1.60	<0.001	1.44–1.76
Place of child’s delivery										
Home										
Public facility		0.88	0.03	0.79–0.99	0.90	0.07	0.80–1.01	0.91	0.10	0.81–1.02
Private facility		0.83	0.02	0.72–0.97	0.84	0.02	0.73–0.98	0.84	0.02	0.72–0.98
Elsewhere		1.04	0.82	0.76–1.41	1.03	0.83	0.76–1.41	1.03	0.85	0.76–1.40
Maternal/caregiver highest educational level										
No education										
Primary		0.83	0.02	0.72–0.97	0.85	0.03	0.73–0.99	0.85	0.03	0.73–0.99
Secondary		0.67	<0.001	0.57–0.77	0.67	<0.001	0.57–0.79	0.67	<0.001	0.57–0.79
Higher		0.44	<0.001	0.36–0.55	0.45	<0.001	0.36–0.56	0.46	<0.001	0.36–0.57
Mother staying with a partner										
Staying with partner										
Staying elsewhere		0.96	0.59	0.82–1.12	0.96	0.64	0.82–1.13	0.96	0.57	0.81–1.12
Mother’s religious status										
Catholic										
Other Christian		1.00	0.97	0.85–1.17	1.01	0.89	0.86–1.19	1.03	0.69	0.88–1.22
Islam		1.15	0.15	0.95–1.40	1.19	0.09	0.98–1.45	1.28	0.02	1.04–1.58
Traditionalist and others		0.94	0.79	0.58–1.51	0.93	0.77	0.58–1.50	0.95	0.83	0.59–1.53
Mother’s anaemia status										
Not anaemic										
Anaemic		1.58	<0.001	1.45–1.72	1.58	<0.001	1.45–1.72	1.58	<0.001	1.45–1.72
Maternal body mass index										
Normal										
Underweight		1.07	0.26	0.95–1.21	1.08	0.23	0.96–1.21	1.08	0.19	0.96–1.22
Overweight		0.76	<0.001	0.68–0.85	0.77	<0.001	0.68–0.86	0.77	<0.001	0.68–0.86
Obese		0.67	<0.001	0.57–0.78	0.68	<0.001	0.58–0.79	0.68	<0.001	0.58–0.79
Paternal work status										
Not working										
Working		1.21	0.14	0.94–1.57	1.21	0.15	0.93–1.56	1.19	0.19	0.92–1.54
Partner education status										
No education										
Primary education		0.90	0.21	0.77–1.06	0.92	0.30	0.78–1.08	0.91	0.27	0.78–1.07
Secondary education		0.82	0.01	0.70–0.95	0.83	0.02	0.72–0.97	0.84	0.02	0.72–0.98
Tertiary education		0.72	<0.001	0.60–0.86	0.73	<0.001	0.61–0.88	0.74	<0.001	0.61–0.89
Household wealth index										
Poorest										
Poorer		0.83	0.02	0.71–0.97	0.87	0.09	0.74–1.02	0.87	0.08	0.74–1.02
Middle		0.60	<0.001	0.51–0.71	0.71	<0.001	0.59–0.85	0.72	<0.001	0.60–0.86
Richer		0.47	<0.001	0.39–0.56	0.59	<0.001	0.48–0.73	0.61	<0.001	0.50–0.76
Richest		0.32	<0.001	0.26–0.39	0.41	<0.001	0.33–0.53	0.43	<0.001	0.34–0.55
Children under 5 slept under a bed net										
No child										
All children		0.90	0.13	0.79–1.03	0.90	0.13	0.78–1.03	0.89	0.10	0.78–1.02
Some children		1.10	0.30	0.92–1.32	1.10	0.31	0.92–1.32	1.10	0.31	0.92–1.32
No net in household		0.89	0.11	0.78–1.02	0.90	0.12	0.78–1.03	0.90	0.13	0.78–1.03
Sex of household head										
Male										
Female		0.93	0.34	0.80–1.08	0.93	0.37	0.81–1.08	0.94	0.38	0.81–1.08
Number of people in household										
2–3										
4–6		0.94	0.42	0.80–1.10	0.94	0.42	0.80–1.10	0.94	0.40	0.80–1.09
7–9		1.00	0.99	0.84–1.19	1.00	0.96	0.83–1.19	0.99	0.92	0.83–1.18
10+		1.18	0.10	0.97–1.43	1.17	0.12	0.96–1.42	1.17	0.12	0.96–1.42
Median community wealth level										
Low										
High					0.73	<0.001	0.62–0.87	0.79	0.01	0.67–0.94
Median proportion of community distance to health facility is no big problem										
Low										
High					0.84	0.01	0.75–0.95	0.85	0.01	0.76–0.96
Media proportion of community maternal education level										
Low										
High					1.05	0.56	0.89–1.25	1.06	0.50	0.89–1.26
State multidimensional poverty index (SMPI)										
Highly deprived										
Above averagely deprived								1.35	0.05	1.01–1.83
Averagely deprived								0.75	0.16	0.50–1.12
Mildly deprived								0.72	0.14	0.47–1.11
Least deprived								0.74	0.23	0.44–1.22
State human development index (HDI)										
Lowest HDI										
Low HDI								1.30	0.06	0.99–1.72
Average HDI								1.31	0.12	0.93–1.86
High HDI								1.47	0.07	0.97–2.22
Highest HDI								1.18	0.48	0.75–1.87
Gender inequality index by state (GII)										
Lowest GII										
Low GII								0.78	0.08	0.60–1.02
Average GII								1.32	0.06	0.99–1.77
High GII								1.01	0.96	0.78–1.29
Highest GII								1.30	0.11	0.94–1.79
Region of residence										
North-central										
North-east								0.68	0.03	0.49–0.96
North-west								1.19	0.36	0.82–1.74
South-east								1.60	<0.001	1.17–2.18
South-south								1.63	<0.001	1.17–2.25
South-west								1.66	<0.001	1.19–2.30
Type of place of residence										
Urban										
Rural								1.29	<0.001	1.13–1.46
Variance components										
Community level variance	0.81	0.31		0.24–0.40	0.31		0.25–0.40	0.30		0.24–0.39
State level variance	0.47	0.08		0.04–0.14	0.09		0.05–0.16	0.02		0.01–0.05
ICC at the community level	0.28	0.11		0.09–0.13	0.11		0.09–0.13	0.09		0.07–0.11
ICC at the state level	0.10	0.02		0.01–0.04	0.02		0.1–0.04	0.004		0.00–0.01
VPC at the community level	0.18	0.08			0.08			0.08		
VPC at the state level	0.10	0.02			0.02			0.004		
MOR at the community level	2.36	1.70		1.60–1.83	1.70		1.61–1.83	1.69		1.60–1.81
MOR at the state level	1.92	1.31		1.21–1.43	1.33		1.24–1.46	1.14		1.10–1.24
AIC	19,609	18,553			18,535			18,519		
BIC	19,638	18,909			18,912			19,027		
Log-likelihood	−9800	−9227			9215			−9189		

AOR: adjusted odds ratios, CI: confidence interval, ICC: intraclass correlation coefficient, VPC: variance partition component, MOR: median odds ratios, AIC: Akaike information criteria, BIC: Bayesian information criteria.

## Data Availability

The data set used in this study is available in MeasureDHS https://dhsprogram.com (accessed on 28 January 2020) and UNDP-Nigeria https://www.undp.org/nigeria/publications/national-human-development-report-2018 (accessed on 3 March 2020).

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
