# Peer review of "The Risk Factors Associated with the Prevalence of Multimorbidity of Anaemia, Malaria, and Malnutrition among Children Aged 6–59 Months in Nigeria"

_ijerph, 2024, doi:10.3390/ijerph21060765_

Round 1
Reviewer 1 Report
Comments and Suggestions for Authors
Page 6 of 25: Table 1: Colum 5: Title should be revised to be Two or more of the diseases.
Author Response
Thank you for the comments made on our paper. Our reply is in the attached documents

Reviewer 2 Report
Comments and Suggestions for Authors
The formatting of the tables in this article needs adjustment and beautification. It is rare for tables in articles to occupy so many pages. If they are too long, they should be placed in an appendix.
The author is advised to provide detailed information about the two datasets or link to relevant resources. For instance, whether the data is from a nationwide census or selected regions, and how the selection was made. The author is advised to provide detailed information on how the outcome variables were collected, such as which tools were used to measure the three outcome variables, and how the measurements were conducted.
The author is advised to provide a detailed description of the collection process for the influencing factors. Were they collected along with the outcome variables? Additionally, how was household wealth measured?
The author is advised to cite a more comprehensive range of references. Currently, there are some literature on anemia that has not been cited.
Author Response
Thank you for the comments made on our paper that will greatly improve the paper. Our reply is in the attached documents

Reviewer 3 Report
Comments and Suggestions for Authors
This study used two nationally representative cross-sectional surveys to investigate what risk factors are associated with the prevalence of multimorbidity among children aged 6 to 59 months in Nigeria. The results can provide reliable evidence for the development of relevant policies. There are still some issues in the manuscript that need to be confirmed by the author.
1. Please explain why categories 2 and 3 were grouped into one and analyzed during the analysis.
2. The author thinks there is no standard procedure to check the proportionality assumptions using a multilevel mixed effect ordered logistic regression model, so the author used a simple recommended approach, Please indicate the applicability of this method to such studies and whether the accuracy of this method has been verified in other literature.
3. The author analyzes the spatial information of two or more diseases by states & FCT and region of residence only through multimorbidity data, However, the relationship and difference between them are not fully explained. Please add more specific information.
Author Response
Thank you for the comments made on our paper, which will greatly improve our paper. Our reply is in the attached documents
